# A Mechanochemical Route for ZnS Nanocrystals, and Batch Sorting along Size Distribution

**DOI:** 10.3390/nano9091325

**Published:** 2019-09-15

**Authors:** Pengfei Hu, Chen Xie, Zhihui Mao, Xue Liang

**Affiliations:** Laboratory for Microstructures, Shanghai University, Shanghai 200444, China; hpf-hqx@shu.edu.cn (P.H.); FHYKong@shu.edu.cn (C.X.); maozhihui@shu.edu.cn (Z.M.)

**Keywords:** nanocrystals, mechanochemical synthesis, solid state vesicles, batch-sorting, size evolution, size-dependent property

## Abstract

The assistances of sodium dodecyl benzene sulfonate (SDBS) and aging treatment were introduced to further improve the room-temperature mechanochemical synthesis of the quantum-sized zinc sulfide (ZnS) nanocrystals. As a result, a green strategy for synthesizing the monodisperse nanocrystals with tunable size and crystallinity was developed, holding convenient, highly efficient and low pollution. Size evolution shows a gradually increasing trend along the aging-temperature. A model that the independent reaction cells constructed by SDBS-wrapped reactant packages (solid state vesicles, SSVs) for the confined growth of ZnS nanocrystals was proposed to access the formation mechanism of ZnS quantized crystal in a solid-state synthesis system. The band gaps and band-edge luminescent emissions of as-prepared ZnS nanocrystals experienced the size-dependent quantum confinement effect, while the trap-state emissions exhibited the lattice integrity-dependence. Furthermore, ZnS quantum-sized nanocrystals with narrower size distribution can be obtained by a batch-sorting process through adjusting the centrifugal speed.

## 1. Introduction

For decades, the study for the quantized semiconductor structures was lastingly set off owing to their fascinating size-, and shape-dependent electronic and optical properties, such as how the band gap of a semiconductor nanostructure increases as the particle size decreases when the dimension of nanocrystallites approach the exciton Bohr radius, due to the quantum-size confinement effect [1,2,3]. As one important member of the II–VI group of semiconductors, zinc sulfide (ZnS, direct band gap of 3.7 eV) motivated the enthusiasm of scientists worldwide. Especially, the ZnS quantum dots (Q-ZnS) and Q-ZnS-based heterostructures have become essential light-harvesting and light-emitting components in modern technology, such as bio-imaging, biosensing, diagnostics, solar cells, and display technologies [4,5,6]. Several methods have been studied to synthesize ZnS quantized nanocrystals and its heterojunction structure. Among them, the liquid phase preparation route is quite popular [7,8,9,10], such as thermolysis of single-source precursors containing zinc and sulfur ligands [10], which generally requires rather complicated procedures including delicate control of surfactant ratios and inert reaction conditions. Therefore, exploring green, convenient, and effective synthesis methods has always been a hot challenge.

On the other hand, it is well known that sodium dodecyl benzene sulfonate (SDBS) is an anionic surfactant and is widely used as a regulator in liquid phase synthesis of nanomaterials [11,12,13,14,15]. In the first case, as a dispersing or stabilizing agent, SDBS can disperse reactants or stabilize products, thus regulating the reaction process or reducing the agglomeration of product particles. This is one of the main starting points for the use of SDBS in the synthesis of nanomaterials in a liquid phase system. Secondly, SDBS can selectively adsorb on the specific crystal plane of the produced nanocrystals, which hinders the growth of the crystal plane and promotes the growth of other crystal planes, thus controlling the morphology of the product. For example, Cao et al. have reported that SDBS selectively adsorbed on the (100) crystal faces of lead sulfide (PbS) nanocrystals, which hindered their growth and promoted the growth of nanocrystals along <111> directions, thus obtaining a perfect PbS cubic frame with six pyramid-shaped hoppers negative toward the center on the (100) facets and eight elongated horns along the <111> directions. However, the above summary of the role of SDBS is based on the liquid-phase reaction system. As far as we know, there are few reports on the role of SDBS in the solid-state reaction system. Studying the regulation of SDBS on products in solid-state reaction synthesis can expand its application and nano-synthesis methods.

The low-temperature solid-state chemical reaction technique, which has the advantages of convenient operation, low cost, and less pollution, has provided a relatively simple and powerful method for the synthesis of nanomaterials [16,17,18,19,20]. However, the crystallinity of nanomaterials synthesized in this way is often not perfect, as well as this, the surface of particles often has defects such as protuberance. In the present work, the SDBS-assisted room-temperature mechanochemical synthesis methodology was developed by following an aging treatment in order to produce quantized structures with improved structure and performance. Happily, the Q-ZnS with tunable size and good lattice integrity has been successfully harvested, avoiding the time/energy-consumption often happening in the solution-based process. At the same time, the influence of temperature to the crystallization degree of Q-ZnS and their size/lattice integrity-dependent optical properties were investigated. Moreover, the multidimensional effects of SDBS on solid-state nanosynthesis, including dispersion, stability and ligands, were proposed and discussed. This strategy expands the room-temperature mechanochemical synthesis methodology for the preparation of nanocrystals.

## 2. Experimental

The Q-ZnS was synthesized in accordance with the procedure illustrated in references [11,12,13,14,15] and following the subsequent aging treatment. Figure 1 illustrates the manipulation for the synthesis of Q-ZnS. Firstly, Zinc acetate dihydrate (Zn(OAc)_2_·2H_2_O) powders were blended with surfactants SDBS at a molar ratio of 5:1 (Zn(OAc)_2_·2H_2_O/additive) and ground together for 10 min at room temperature (Grinding operation: holding the handle of the pestle and pressing on the reagent in the pestle to rotate at about 60–80 revolutions per minute (rpm)). Then, the sodium sulfide hydrate (Na_2_S·9H_2_O) was added with a molar ratio of 1:1 between Zn(OAc)_2_·2H_2_O and Na_2_S·9H_2_O. The mixture was ground for about 30 min to obtain a masterbatch. In the subsequent process, the masterbatch was collected into a vessel and preserved with oil baths at different temperatures (50 °C, 100 °C, and 150 °C, respectively) for 12 h. Finally, the aged masterbatch was centrifugation washed with deionized water (three times) and absolute ethanol (one time) respectively, and dried at 60 °C to obtain the desired product. According to the aging temperature, the samples were named Q-ZnS-I, Q-ZnS-I′, and Q-ZnS-I″, respectively (Q-ZnS-I: 50 °C, Q-ZnS-I′: 100 °C, Q-ZnS-I″: 150 °C). Several batches of ZnS quantum dots with different particle size distributions will be obtained by adjusting the rotational speed of centrifuge (3000 and 4000 rpm, respectively) from the Q-ZnS-I′. Herein, the redispersed sample, Q-ZnS-I′, was centrifuged at 3000 and 4000 rpm respectively, and supernatant and precipitation were separated. The supernatants were labeled Q-ZnS-I′-i and Q-ZnS-I′-ii, respectively.

The crystalline structures of the products were analyzed by a powder X-ray diffractometer (XRD, MXP18AHF, MAC Science Co. Ltd., Japan) with Cu-Kα radiation (*λ* = 0.154056 nm). The fourier transform infrared spectroscopy (FT-IR) test was carried out by the pressing plate method (compressing mixed powder of sample and potassium bromide into tablets) on a Thermo Nicolet Avatar 370 FT-IR spectrometer (Welltech Scientific Inc., Chantilly, VA, USA) at a resolution of 4 cm^−1^ and 32 scans. Subsequently, the samples were redispersed in deionized water to get a clear solution for the following characterization, respectively. The morphologies, microstructures, and crystal lattice of the Q-ZnS were characterized by transmission electron microscopy (TEM, JEM-2010F at 200 KV, JEOL Ltd., Tokyo, Japan) and scanning electron microscope (SEM, JSM-7500F, JEOL Ltd., Tokyo, Japan). The Ultraviolet-Visible (UV-Vis) and Photoluminescence (PL) spectra were recorded on a HITACHI U-3310 ultraviolet-visible spectrophotometer (Hitachi Production Co., Ltd., Tokyo, Japan) and an Edinburgh FS5 photoluminescence spectrophotometer (Edinburgh Instruments Ltd., Edinburgh, UK) with the above redispersed solution at room temperature, respectively.

## 3. Results and Discussion

### 3.1. Crystal Structures of Zinc Sulfide (ZnS) Nanocrystals

The X-ray diffraction (XRD) pattern of as-prepared Q-ZnS is shown in Figure 2. Three characteristic diffraction peaks at 2θ values of about 29.14/28.82/28.91, 48.23/48.29/48.16, and 57.11/56.8/56.9 to samples I, I′, and I″ can be indexed to the crystal planes of (1 1 1), (2 2 0), and (3 1 1) of cubic phase ZnS respectively, which matched with the reported data very well (JCPDS card file No. 65-9585, standard values of three diffraction peaks: 28.56, 47.50, and 56.37, respectively). The results showed that the diffraction peak shifts to the right relative to the standard sample in varying degrees, suggesting that the lattice shrinkage exists in the sample particles. At the same time, the diffraction peaks show obvious broadening, suggesting that the sizes of the nanocrystals are small, and these small grains will cause lattice shrinkage. On the other hand, the comparison of the diffraction peaks of samples I, I′, and I″ showed that the intensity of the diffraction peaks enhanced along the increase of aging temperature, implying that the crystallinity of nanocrystals in the Q-ZnS was improved with the increase of the aging temperature.

Figure 3 shows the TEM characterizations and high resolution transmission electron microscope (HRTEM) resolutions of representative nanocrystals for respective Q-ZnS (Q-ZnS-I, Q-ZnS-I′, and Q-ZnS-I″). It is shown that most of the Q-ZnS nanocrystals are the mature objects with quasi-spherical shape, relative clear-cut lattice fringe patterns, and smoother surface. The size of the nanocrystals increases from Q-ZnS-I to Q-ZnS-I″, at the same time, the integrity of crystallization of the nanocrystals is also gradually improved in the same order (Figure 3a,a′,b,b′,c,c′). On the whole, with the increase of aging temperature, the nanocrystals gradually grow up, displaying the grain size increases and the lattice integrity improvement. To sum up, it is obvious that the aging temperature is another important regulation means for the size, shape and lattice integrity of nanocrystals besides the additive, in the low-temperature, solid-state chemical synthesis strategy. It is worthwhile to note that some nanocrystals in Q-ZnS-I″ generated zigzag twinning structures containing a large number of stacking fault and dislocation which are similar to previous reports [21], and several nanocrystals formed agglomeration (Appendix A).

Now, in the present room-temperature solid-state reaction system, we proposed that SDBS plays the role of dispersant and stabilizer like in the liquid-phase reaction system. A simple scheme to visualize the formation mechanism of ZnS nanocrystals is given in Scheme 1. Firstly, after grinding, the zinc acetate powders are evenly dispersed into SDBS powder, forming many SDBS-wrapped zinc acetate packages (solid state vesicles, SDBS-Zn-SSVs). These SDBS-Zn-SSVs were characterized through SEM and EDS-mapping technologies (Bottom of Scheme 1). It is shown that the zinc species accompanied with oxygen occupied the central area of SDBS-Zn-SSV, while sulfur is more uniformly dispersed throughout the vesicle, not showing an obvious enrichment zone. Additionally, since the substrate carrying sample is conductive tape (an organic carbon material), the mapping of carbon elements loses its reference value and is not presented. Each SSV can act as an independent reaction cell with quantitative zinc acetate. When the reactant sodium sulfide is added, it will diffuse into the cell, and the reaction of forming ZnS (equation in Scheme 1) occurs in this limited space. Subsequently, as-synthesized ZnS will be confined to grow into nanocrystals with certain shape and size in this space. That is to say, every SDBS-Zn-SSV is a confined reaction and growth space. In addition, one end of the SDBS molecule is a long chain alkyl and the other end is a sulfonic acid group containing an oxygen atom which can provide electronic coordination. So, the as-prepared ZnS nanocrystals were stabilized by SDBS through a coordination effect and long-chain alkyl coverage, suppressing the agglomeration of nanocrystals. Here, the stabilization of SDBS on the ZnS nanocrystals can be identified by FT-IR characterization. The FT-IR spectra (401–4000 cm^−1^) of Q-ZnS-I′ and SDBS are compared in Figure 4. It is clearly displayed that the Q-ZnS-I′ sample shows the characteristic peaks of sulfonate at around 691–559 cm^−1^, 841 cm^−1^, and 1210–1026 cm^−1^ (denoted by the red arrows in Figure 4) and presents an obvious shift and weakening compared with that of SDBS, suggesting that SDBS is coated on the surface of nanocrystals. Ultimately, monodisperse ZnS nanocrystals can be harvested. It is conceivable that, also confirmed by parallel experiment (Appendix A) without SDBS, zinc acetate and sodium sulfide will contact directly and react rapidly, reluctantly getting large ZnS aggregates (aging at 100 °C). This may be due to the lack of dispersion and stabilization of SDBS in this case.

On the other hand, based on the above model, it can be deduced that the size of ZnS nanocrystals would be controlled by the size of SDBS-wrapped reactant packages. In the present case, the size distribution of ZnS nanocrystals is wide, which can be attributed to the difference in the size of the SDBS-wrapped reactant packages. Furthermore, the zigzag twinning structure and slight agglomeration in sample Q-ZnS-I″ may be due to some SDBS-wrapped packages being broken in the higher temperature. This leads to interface contact and fusion of ZnS nanocrystals, resulting in the dislocation and aggregation.

### 3.2. Optical Properties of ZnS Nanocrystals

#### 3.2.1. Ultraviolet-Visible (UV-Vis) Spectral Characteristics

It is well known that the semiconductor nanocrystals will show strong size-dependent optical properties when the size of crystallite has been reduced to about the exciton Bohr radius scale. The UV-Vis spectra of nanoscale particles can provide some readable optical information, containing the position and intensity of absorption peaks and the band gap values can be estimated by them. In addition, some microstructural information of the nanocrystals can be deduced by the UV-Vis spectra, especially the estimation of reasonable particle size. Several models aiming to estimate the size of Q-ZnS particles, such as the finite-depth model and the tight-binding model [22,23], were respectively proposed by Nosaka and Koster to estimate the size of the nanocrystal by calibrating the curve of the size against the peak wavelength of the absorption spectra.

Figure 5a shows the UV-Vis absorption spectra of the Q-ZnS-I, I′, and I″. They all show a superimposition pattern which consists of a broad absorption band locating in the far ultraviolet and violet regions and a long tail after 350 nm, respectively. Overall, the absorption band from the high energy region to the low energy region displays a slope absorption curve with a relatively well-defined peak around 260 nm, 265 nm, and 281 nm for Q-ZnS-I, Q-ZnS-I′, and Q-ZnS-I″, respectively. From the comparison of the UV-Vis absorption and TEM characterization, their absorption peak shifted gradually with the increase of particle size. In general, the band gap values can be obtained from the UV-Vis absorption spectra by plotting (αhν)^2^ versus (hν), and then extrapolating the linear region of (αhν)^2^ to the intersection of the energy axis (hν). Using this method, the band gap values of Q-ZnS-I, Q-ZnS-I′, and Q-ZnS-I″ were estimated to be 4.85, 4.76, and 4.58 eV, respectively (Figure 5a). Compared with their bulk counterparts (344 nm of absorption peak and 3.68 eV of band gap value), these quantum-sized ZnS nanocrystals exhibited a big blue shift of the absorption band, indicating that the as-prepared Q-ZnS-I, I′, and I″ offer a strong quantum confinement effect.

#### 3.2.2. Photoluminescence Studies

Photoluminescence is another important optical performance for the semiconductor nanostructures [24,25]. The luminescence is a process of light energy absorption, transformation and re-emitting. Firstly, the luminescent materials in the ground state can be excited to an ‘excited state’ under the irradiation of the light with a specific wavelength, in which the electrons jump to a higher energy level, and it is a higher energy state than the ground state. Subsequently, these excited electrons undergo radiative deexcitation and return to the ground state or lower energy level at the same time to emit the same or different from the original excitation wavelength. Theoretically, luminescent emission includes band-edge emission and trap-state emission, the former is derived from the recombination of the exciton, which belongs to the intrinsic luminescence. On the other hand, the latter stems from the separation of the carrier (electron and hole) in the trap from the trap and the combination of them with the luminescent center. It is divided into shallow-trap and deep-trap emissions according to the difference of electron levels [26]. Generally, the defects which mostly include vacancies and interstitials of semiconductors are the main original place of the trap-stated PL signals. The defects can be categorized as either shallow or deep level defect states. Shallow level traps, which are more spatially delocalized and lie near the conduction band or valance band edge, are more likely to participate in the radiative recombination. The emission occurs when a trapped electron recombines with a hole in the valence band or in some acceptor level such as zinc vacancies [27]. On the other hand, deep level traps tend to have a tendency to undergo non-radiative recombination by emitting phonons [26].

Photoluminescence spectra of Q-ZnS-I showed a broad violet emission peak centered at 384 nm accompanied by the broad asymmetry shoulder at 427 nm, and smaller blue emission peaks at 497 nm, respectively (Figure 5b). Because of the wide size distribution of the nanocrystals, its band-edge emission and trap-state emission peaks overlap, showing a broad hilly emission spectrum. Meanwhile, the intensities of emission patterns presented a high and low value crisscross layout, which may be attributed to the difference of the number of respective nanocrystals with corresponding size and the structural reasons such as the density of defects. We attribute the 384 nm peak to a band-edge emission. The shoulder centered at 427 nm is assigned to the overlapping peaks involving electronic transitions from the conduction band to interstitial sulfur and interstitial zinc to the valance band. In addition, the peak at 497 nm is assigned to the recombination of electrons at sulfur vacancies to the holes at zinc vacancies. Compared to the sample Q-ZnS-I, the samples Q-ZnS-I′ and Q-ZnS-I″ displayed similar fluorescence patterns, but the excitonic emission peaks gradually displayed a red shift, and the intensity increased along with the heightening of the aging temperature. These evolution trends are the reflection of the structural changes of the nanocrystals. The grain size of the nanocrystals increases with the increase of the aging temperature, which causes the band edge emission red shift. On the other hand, the lattice integrity of the nanocrystals increases with the increase of the aging temperature, which leads to the enhancement of fluorescence. In addition, it is worth noticing that the photoluminescence peaks coming from the sulfur vacancies or interstitial zinc are similar, emitting at about same wavenumber, implying the trap-state emission is lattice integrity-dependent (the lattice integrities of three samples are not a big difference).

### 3.3. Batch-Sorting of ZnS Nanocrystals

The batch of ZnS nanocrystals with different size distributions obtained through the batch-sorting process was collected and characterized by TEM, respectively. Figure 6a,a′,b,b′ displayed the TEM pictures and size distribution plots of the Q-ZnS-I′-i (3000 rpm) and Q-ZnS-I′-ii (4000 rpm), respectively. It is shown that the particle size of ZnS nanocrystals in supernatant obtained at low centrifugal speed presented a size distribution of about 1–8 nm, while that at higher rotational speed presented a narrower distribution of about 1–5.5 nm. Meanwhile, the size of the main nanocrystals in Q-ZnS-I′-ii is smaller than that in Q-ZnS-I′-i. In brief, as to the centrifugal sorting of nanocrystals in suspension, the size distribution of nanocrystals in supernatant obtained at higher centrifugal speed is narrower than that at lower centrifugal speed, that is to say, the sorting effect is better at high speed than that at low speed. A HRTEM resolution of a single nanocrystal clearly revealed high crystallinity of ZnS nanocrystals.

Figure 6c shows the representative UV-Vis absorption and fluorescence spectra of ZnS nanocrystals of Q-ZnS-I′-ii. The monochromaticity of these two spectra is much better than that of products without centrifugal sorting. The absorption spectrum shows a broad excitation peak at about 300 nm. This may be due to a slightly wider size distribution of ZnS nanocrystals of Q-ZnS-I′-ii than an ideal sample with single size distribution. The trend of the absorption curve may correspond to the size distribution of the sample (Figure 6b′). The fluorescence spectrum displays two distinct and independent characteristic peaks at 349 and 457 nm, respectively. The peak at 349 nm has been assigned to band edge emission of ZnS nanocrystals. The fluorescence emission at 457 nm is attributed to the trap states emission of ZnS nanocrystals with larger Stoke displacement and stronger emission intensity than the band-edge emission at 349 nm. The above characteristics of fluorescence emission peaks reflect the small size and narrow distribution of this batch-sorting sample, and the high defect density of the small size nanocrystals possibly due to their immaturity. The fluorescence quantum yield (FLQY) of the sample Q-ZnS-I′-ii was 37.32% (L-tyrosine as reference), which is at the high level of FLQY of ZnS quantum dots or its derivatives reported in the literature [28,29]. Among them, quantum dots prepared by the liquid-phase pathway are the main ones. Therefore, this work provides a new green scheme for preparing ZnS quantum dots with excellent properties. The current work emphasizes again that the synthesis of quantum dot nanocrystals with characteristic emission is very important for later applications. Now, the synthesis of quantum dots with characteristic strong emission has become one of the key tasks, and related work is under way.

## 4. Conclusions

In summary, quantum-sized ZnS nanocrystals with good lattice integrity were obtained through SDBS-assisted room-temperature solid-state reaction methodology attaching aging-treatment. The size-control can be achieved through using different aging temperatures, showing the gradual increase along with the increase of aging temperature. Based on the hypothesis of an independent reaction cell created by SDBS-Zn-SSVs, the synthesis mechanism of monodisperse ZnS nanocrystals is discussed. The as-prepared quantum-sized ZnS nanocrystals exhibited the quantum confinement effect and well-defined size-dependence of UV-vis absorption. The band-edge PL emission showed the mostly size-dependent excitonic emission feature, while the trap-state PL experienced the lattice integrity-dependence. Batch-sorting with controlling centrifugal speed is used to obtain nanocrystals with narrower size distribution, which exhibit excellent optical properties.

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
