# Peer review of "A Mechanochemical Route for ZnS Nanocrystals, and Batch Sorting along Size Distribution"

_nanomaterials, 2019, doi:10.3390/nano9091325_

Round 1

Reviewer 1 Report

The paper could be of interest for the readers of Nanomaterials but it requires major revisions before publishing. English form and spelling should be revised by a native speaker. The mechanochemical synthesis of solid compound is an alternative, often green, procedure which can lead to better products compared to the wet chemical ones (cfr. L. Russo, F. Colangelo, R. Cioffi, I. Rea, L. De Stefano, “A Mechanochemical
Approach to Porous Silicon Nanoparticles Fabrication” Materials 2011, 4, 1023-1033). Authors should strongly improve the Experimental section by reporting all details about the samples preparation and the techniques used in their characterization. For example, FTIR analysis have been performed by casting ZnS powders in KBr or by using ATR technique? PL has been measured on solid or on powders dispersed in solution? XRD usually needs grams of material to record a spectrum: how did authors managed this point?

Authors claim about a model of reaction for the formation of ZnS nanoparticles but they only give a qualitative scheme (Scheme 1) without any quantitative or theoretical reaction that could explain the formation of the semiconductor nanoparticles. A solid state reaction must obey the rule of general chemistry so that authors should propose a scheme of reaction between the constituents of their mixture. The mechanical grinding and the temperature curing give the energy for ZnS nanoparticle formation but a model of reaction should be given. Which is the role of the surfactant: is only a stabilizing agent or it takes part in the chemical reaction? Which is the yield of the process, or in other world, how much material can be fabricated by this method in a day/week? The stability of the particles at acid and basic value of pH should be investigated in order to understand the nature of the aggregation.

In PL characterization it is not reported which is the excitation wavelength used so that it is hard to understand the spectra explanation reported. How do authors justify the big difference in gap values between the measured ones and the bulk ZnS?  

Batch-sorted nanoparticles seem to be much better when compared to the other: could be the process improved in order to get only these good ones?

Author Response

Comments and Suggestions for Authors and our responses:

The paper could be of interest for the readers of Nanomaterials but it requires major revisions before publishing. English form and spelling should be revised by a native speaker. The mechanochemical synthesis of solid compound is an alternative, often green, procedure which can lead to better products compared to the wet chemical ones (cfr. L. Russo, F. Colangelo, R. Cioffi, I. Rea, L. De Stefano, “A Mechanochemical Approach to Porous Silicon Nanoparticles Fabrication” Materials 2011, 4, 1023-1033). Authors should strongly improve the Experimental section by reporting all details about the samples preparation and the techniques used in their characterization. For example, FTIR analysis have been performed by casting ZnS powders in KBr or by using ATR technique? PL has been measured on solid or on powders dispersed in solution? XRD usually needs grams of material to record a spectrum: how did authors managed this point?

ANASWER: Dear reviewers, thank you very much for your professional comments. Blue font is used in the revised part of the text for highlighting.We have carefully revised the manuscript throughout and weighed every sentence. In addition, we have invited Professor Sajid Bashir (Sajid.Liu@tamuk.edu), who worked at Texas A&M University-Kingsville, to polish the article.

1) Firstly, we add the literature of mechanochemical synthesis to the introduction of research background. On page 12, in line 310, the literature “L. Russo, F. Colangelo, R. Cioffi, I. Rea, L. De Stefano, “A Mechanochemical Approach to Porous Silicon Nanoparticles Fabrication” Materials 2011, 4, 1023-1033).” has been added.

2) Secondly, we have carefully revised the experimental part of this paper, including the samples preparation and the techniques used in their characterization.

On page 2, in lines 82-88, the paragraph for the characterization has been modified “The crystalline structures of the products were analyzed by a powder X-ray diffractometer (XRD, MXP18AHF, MAC) with Cu-Kα radiation (λ = 0.154056 nm). The FT-IR test was carried out by pressing plate method (compressing mixed powder of sample and potassium bromide into tablets) on a Thermo Nicolet Avatar 370 FT-IR spectrometer at a resolution of 4 cm-1 and 32 scans. Subsequently, the samples were redispersed in deionized water to get a clear solution for following characterization, respectively. The morphologies, microstructures, and crystal lattice of the Q-ZnS were characterized by transmission electron microscopy (TEM, JEM-2010F at 200 KV). The Ultraviolet-Visible (UV-Vis) and Photoluminescence (PL) spectra were recorded on HITACHI U-3310 ultraviolet-visible spectrophotometer and Edinburgh FS5 photoluminescence spectrophotometer with above redispersed solution at room temperature, respectively.”. It includes the pressing plate method for FIIR test and the solution sample for ultraviolet and fluorescence detection.

3) Third, the product yield of this experimental scheme is high (up to 90%, or even higher), and it is easy to reach the amount required for XRD test. At the same time, the repeatability of this experiment is good, and the yield can also be increased by increasing batches.

Authors claim about a model of reaction for the formation of ZnS nanoparticles but they only give a qualitative scheme (Scheme 1) without any quantitative or theoretical reaction that could explain the formation of the semiconductor nanoparticles. A solid state reaction must obey the rule of general chemistry so that authors should propose a scheme of reaction between the constituents of their mixture. The mechanical grinding and the temperature curing give the energy for ZnS nanoparticle formation but a model of reaction should be given. Which is the role of the surfactant: is only a stabilizing agent or it takes part in the chemical reaction? Which is the yield of the process, or in other world, how much material can be fabricated by this method in a day/week? The stability of the particles at acid and basic value of pH should be investigated in order to understand the nature of the aggregation.

ANASWER: Dear reviewers, in this reaction system, zinc acetate and sodium sulfide are prone to metathetical reaction to produce zinc sulfide (ZnS), which can be directly proved by XRD data. Several parallel experiments have proved that the yield of zinc sulfide can reach more than 90% by the reaction of zinc acetate and sodium sulfide. Now, we have revised the relevant parts of the article.

On page 6, in lines 132-133, the sentence “, causing subsequent  reactions and growth  of ZnS were  limited only in this room” has been modified to “When the reactant sodium sulfide is added, it will diffuse into the cell, and the reaction of forming ZnS (equation in Scheme) occurs in this limited space. Subsequently, as-synthesized ZnS will be confined to grow into nanoparticles with certain shape and size in this space.”.

In the process of mechanochemical reaction, mechanical grinding provides direct power for the diffusion of reactants and promotes the occurrence of reactions. Moreover, temperature aging can provide a driving force for the growth of nanoparticles. But in the solid phase system, this driving force is different from the Ostwald ripening effect in the liquid phase system. These two issues are very important for the study of the mechanism of mechanochemical reaction synthesis, and they are in progress, not as the theme of this work, Dear reviewer.

In the present room-temperature solid-state reaction system, we proposed that SDBS plays the role of dispersant and stabilizer. Firstly, SDBS acts as dispersant in the grinding of zinc acetate and SDBS, avoiding the rapid reaction of Zn(OAc)2 and Na2S after adding Na2S with incontrollable speed. Secondly, SDBS can stabilize the surface of zinc sulfide particles through coordination bonds and other modes, thus stabilizing the nanoparticles.

As mentioned above, the yield of ZnS can reach more than 90% by the reaction of Zn(OAc)2 and Na2S in the present experiment. At present, this work is in the laboratory synthetic experiment stage, has not reached the pilot-scale and industrial production stage, no textual research on daily or weekly production.

Because the Ksp of ZnS (2.93×10-25) is much smaller than that of Zn(OH)2 (1.2×10-17), ZnS is relatively stable in alkaline solution. But, sulphur in ZnS can bind with hydrogen ions ionized by acid to form volatile H2S, which leads to the gradual dissolution of ZnS in acid. The work of this paper is to synthesize samples in the solid state, which does not involve the PH value of the solution (although water is added during washing, the product ZnS nanoparticles or aggregations have been formed), so this point is not discussed in this paper.

In PL characterization it is not reported which is the excitation wavelength used so that it is hard to understand the spectra explanation reported. How do authors justify the big difference in gap values between the measured ones and the bulk ZnS?  

ANASWER: The excitation wavelength in PL characterization for Q-ZnS-Iʹ-ii is 275 nm (shown in Fig 6c), while the excitation wavelengths in PL characterization for Q-ZnS-I, Iʹ, and I'' also are based on absorption peaks, respectively.

Due to the small particle size (less than Bohr radius), the ultraviolet absorption peaks of as-synthesized ZnS nanoparticles are blue-shifted (344 nm absorption wavelength of bulk ZnS) and the band gap values are larger than 3.68 eV of bulk ZnS (4.85, 4.76, and 4.58 eV for Q-ZnS-I, ()Q-ZnS-Iʹ, and Q-ZnS-Iʹʹ, respectively).

On page 8, in lines 180-181, the sentence “Using this method, the band gap values were estimated  to  be  4.85,  4.76,  and  4.58  eV  for  Q-ZnS-I,  b,  and  c,  respectively  (Fig.  5a).” has been changed to “Using this method, the band gap values of Q-ZnS-I, Q-ZnS-Iʹ, and Q-ZnS-Iʹʹ were estimated to be 4.85, 4.76, and 4.58 eV, respectively (Fig. 5a).”.

On page 8, in line 182, the phrase “344 nm of absorption peak and 3.68 eV of band gap value” has been added.

Batch-sorted nanoparticles seem to be much better when compared to the other: could be the process improved in order to get only these good ones?

ANASWER: Dear reviewers, we are trying to explore the relevant processes in order to get the product with desired size and size distribution, including the establishment of the function relationship between centrifuge speed and particle size distribution, and the concentration of supernatant followed by secondary batch-sorting.

Reviewer 2 Report

Hu et al. present an interesting study on the solid state synthesis of ZnS nanocrystals using the surfactant sodium dodecyl benzene sulfonate (SDBS) as an additive in the process. The paper is generally easy to follow, and they provide some interesting observations on this system. However, there are some problems in my opinion.

Firstly, I there should be some comparison made with ZnS made by solution-phase approaches (comparing with results from the literature would be fine). For example, PLQY? In isolation, it’s hard to get a sense of how this method compares.

Secondly, I think there should be some discussion of the structure of SDBS and the binding mechanism with the ZnS nanocrystals. Or are they simply making solid state vesicles in which the nanoparticles grow? I haven't come across such solid state vesicles before. Could the authors provide some literature references to these?

Thirdly, I feel the spectra presented in Figure 6c, as the surrounding discussion, is not very convincing (see my comments below).

Further general comments:

In the condition without SDBS that was used as a control (Fig S2), what was the temperature of ageing?

The PLQY was not measured, so the spectra shown in Fig 5b only serve as a comparison between conditions, but with no relation to actual QY. Also the spectra in Fig 5b could be quantised somehow.

The original emission spectra are so wide and structured, nothing like the characteristic sharp emission of solution-synthesized QDs. Even after size selective purification there is an abundance of trap states. I wonder how realistic it is to consider this a viable method of making useful QDs. Could the authors justify this in the manuscript?

The manuscript title in the SI is different from the main paper.

Bohr spelt incorrectly in the intro.

Maybe add the chemical structure of SDBS into one of the figures. I think it’s informative to the reader to see this.

How was the grinding performed? There appears to be no detail given. Fig 1 suggests pestle and mortar?

Why was the zinc precursor ground with the SDBA before adding the sulfur precursor? What’s the rationale for this?

What solvent was used during the centrifugation washing step? How was the batch sorting performed? The description lacks detail.

In the sentence “In this case, the size distribution of ZnS nanocrystals is still slightly wider, which can be attributed to the difference in the size of SDBS-wrapped reactant package”, it is not clear what ‘wider’ is referring to, making the comparison hard to understand.

The terminology referring to the three different samples gets confused e.g. “…the samples Q-ZnS-Iʹ and c displayed…“, also in the materials and methods the labelling refers to a, b, c but it’s not obvious why.

“The absorption spectrum shows an excitonic peak at around 269 nm, accompanied by a steamed bun tail.” The UVvis spectrum looks incorrect. I would personally interpret this as a wide excitionic peak around 300 nm, with the peak at ca. 275 nm and the subsequent drop to near zero below that an indication that the cuvette was not suitable for measurements below ca. 290 nm. Were the cuvette and spectrometer the same as the one used in Figure 5a?

Comparing Figure 5b with Figure 6c, there is a very large discrepancy in the spectra for the I’ sample. Figure 5b shows near zero emission intensity at 350 nm, whereas Figure 6c has a distinct peak at 350 nm which is ascribed to band edge emission. Further, the supposed trap state emission at 460 nm in Figure 6c is far higher than the band edge emission, whereas in Figure 5b the trap state is lower in intensity, and also at a much different wavelength position. It is quite confusing, and I think this section needs to some much more careful explanation.

Also, the PL in Figure 6b, where is the baseline? It seems to be much higher than it should be?

Author Response

Comments and Suggestions for Authors and our responses:

Hu et al. present an interesting study on the solid state synthesis of ZnS nanocrystals using the surfactant sodium dodecyl benzene sulfonate (SDBS) as an additive in the process. The paper is generally easy to follow, and they provide some interesting observations on this system. However, there are some problems in my opinion.

Firstly, I there should be some comparison made with ZnS made by solution-phase approaches (comparing with results from the literature would be fine). For example, PLQY? In isolation, it’s hard to get a sense of how this method compares.

ANASWER: Dear reviewers, thank you very much for your professional comments. Blue font is used in the revised part of the text for highlighting.

Using L-tyrosine as a reference, the fluorescence quantum yield of the sample Q-ZnS-Iʹ-ii was determined and calculated to reach 37.32% (the fluorescence integral area of the sample is 44213261.7, and the maximum absorbance is 0.0478). By introducing literature comparison (literatures 27-29), it is pointed out that the mechanochemical synthesis scheme designed in this work is a new and effective green method for synthesizing ZnS quantum dots.

On page 10, in line 239, the sentences “The fluorescence quantum yield (FLQY) of the sample Q-ZnS-Iʹ-ii is 37.32% (L-tyrosine as reference), which is at the high level of FLQY of ZnS quantum dots or its derivatives reported in literature [27, 28]. Among them, quantum dots prepared by liquid-phase pathway are the main ones. Therefore, this work provides a new green scheme for preparing ZnS quantum dots with excellent properties. Current work emphasizes again that the synthesis of quantum dot nanocrystals with characteristic emission is very important for later applications. Now, the synthesis of quantum dots with characteristic strong emission has become one of the key tasks, and related work is under way.” have been added.

Secondly, I think there should be some discussion of the structure of SDBS and the binding mechanism with the ZnS nanocrystals. Or are they simply making solid state vesicles in which the nanoparticles grow? I haven't come across such solid state vesicles before. Could the authors provide some literature references to these?

ANASWER: Dear reviewer, in the present room-temperature solid-state reaction system, we proposed that SDBS plays the role of dispersant and stabilizer. Firstly, SDBS acts as dispersant by forming solid state vesicles in the grinding of zinc acetate and SDBS, avoiding the rapid reaction of Zn(OAc)2 and Na2S after adding Na2S with incontrollable speed. It provides a confined growth space for nanocrystalline particles. The related solid state vesicles have not been reported in the literature. We inferred this structure by SEM and EDS-mapping characterizations (Scheme 1).

Secondly, SDBS is an anionic surfactant which has one end of a long chain alkyl and other end of sulfonic acid group. So, the ZnS nanocrystals can be stabilized by SDBS through coordination effect and long-chain alkyl coverage, suppressing the agglomeration of nanocrystals.

  On page 6, in line 131, the sentences “(solid state vesicles, SDBS-Zn-SSVs). These SDBS-Zn-SSVs were characterized through SEM and EDS-mapping technologies (Bottom of Scheme 1). It is shown that the zinc species accompanied with oxygen occupied the central area of SDBS-Zn-SSV, while sulfur is more uniformly dispersed throughout the vesicle, not showing obvious enrichment zone. Additionally, since substrate carrying sample is conductive tape (an organic carbon material), the mapping of carbon elements loses its reference value and is not presented.” have been added to describe the solid state vesicles.

On page 6, in lines 133-135, the sentences “At the same time, the as-prepared ZnS nanocrystals were stabilized by SDBS through coordination effect, suppressing  the  agglomeration  of  nanocrystals.” has been modified to “In addition, one end of SDBS molecule is a long chain alkyl and the other end is sulfonic acid group containing oxygen atom which can provide electronic coordination. So, the as-prepared ZnS nanocrystals were stabilized by SDBS through coordination effect and long-chain alkyl coverage, suppressing the agglomeration of nanocrystals.”.

Thirdly, I feel the spectra presented in Figure 6c, as the surrounding discussion, is not very convincing (see my comments below).

ANASWER: Dear reviewer, relevant responses can be found later. Thank you!

Further general comments:

In the condition without SDBS that was used as a control (Fig S2), what was the temperature of ageing?

ANASWER: We have added aging temperature here. On page 6, in line 142, the “(aging at 100 oC)” has been added after the sentence ‘……, reluctantly getting large ZnS aggregates’.

The PLQY was not measured, so the spectra shown in Fig 5b only serve as a comparison between conditions, but with no relation to actual QY. Also the spectra in Fig 5b could be quantised somehow.

ANASWER: Dear reviewer, because the fluorescence emission monochromaticities of samples Q-ZnS-I, Iʹ, and I'' are poor, the measurements for their PLQY were not carried out. The spectra shown in Fig 5b only serve as a comparison of preparation conditions for the QDs.

The original emission spectra are so wide and structured, nothing like the characteristic sharp emission of solution-synthesized QDs. Even after size selective purification there is an abundance of trap states. I wonder how realistic it is to consider this a viable method of making useful QDs. Could the authors justify this in the manuscript?

ANASWER: Dear reviewer, the synthesis of the QDs with characteristic sharp emission is one of the following key tasks. Firstly, we are trying to explore the relevant processes in order to get the product with desired size and size distribution, including the establishment of the function relationship between centrifuge speed and particle size distribution, and the concentration of supernatant followed by secondary batch-sorting. Secondly, measures to promote the maturation of QDs are being explored, including the regulation of aging temperature and time, and surface modification of QDs.

 On page 9, in line 239, the sentence “Now, the synthesis of quantum dots with characteristic strong emission has become one of the key tasks, and related work is under way.” has been added.

The manuscript title in the SI is different from the main paper.

ANASWER: Dear reviewer, now, the title of the manuscript in the main paper and SI is unified as “A mechanochemical route for ZnS nanocrystals, and batch sorting along size distribution”.

Bohr spelt incorrectly in the intro.

ANASWER:  On page 1, in line 27, this has been amended to “Bohr”.

Maybe add the chemical structure of SDBS into one of the figures. I think it’s informative to the reader to see this.

ANASWER: Dear reviewer, we have added the skeleton symbol chemical structure of SDBS into the Scheme 1. (On page 6, in line 144)

How was the grinding performed? There appears to be no detail given. Fig 1 suggests pestle and mortar?

ANASWER: On page 2, in line 72, the sentence “(Grinding operation: holding the handle of the pestle and pressing on the reagent in the pestle to rotate at about 60-80 revolutions per minute (rpm)).” has been added to describe the operation of grinding. Moreover, figure 1 shows the rod-shaped pestle and the container-shaped pestle, dear reviewer.

Why was the zinc precursor ground with the SDBA before adding the sulfur precursor? What’s the rationale for this?

ANASWER: In the present room-temperature solid-state reaction system, we proposed that one of the functions of SDBS is to act as a dispersant. Grinding Zn(OAc)2 with SDBS before adding the Na2S can avoid the rapid reaction of Zn(OAc)2 and Na2S after adding Na2S with incontrollable speed, and agglomeration of particles. This can partly regulate the growth of ZnS.

What solvent was used during the centrifugation washing step? How was the batch sorting performed? The description lacks detail.

ANASWER: On page 2, in lines 75-76, the sentence “Finally, the aged masterbatch was centrifugation washed several times and dried at 60 oC to obtain the desired product.” has been modified to “Finally, the aged masterbatch was centrifugation washed with deionized water (three times) and absolute ethanol (one time) respectively, and dried at 60 oC to obtain the desired product.”.  

Moreover, on page 2, in lines 78-81, the descriptions “Several batches of ZnS quantum dots with different particle size distributions were separated by adjusting the rotational speed of centrifuge (3000 and 4000 rpm (revolutions per minute), respectively) from the Q-ZnS-Iʹ. The supernatant obtained at 3000 and 4000 rpm of the sample is recorded as the Q-ZnS-Iʹ-i and Q-ZnS-Iʹ-ii.” have been modified to “Several batches of ZnS quantum dots with different particle size distributions will be obtained by adjusting the rotational speed of centrifuge (3000 and 4000 rpm, respectively) from the Q-ZnS-Iʹ. Herein, the redispersed sample Q-ZnS-Iʹ was centrifuged at 3000 and 4000 rpm, respectively, and supernatant and precipitation were separated. The supernatants are labeled Q-ZnS-Iʹ-i and Q-ZnS-Iʹ-ii, respectively.” to describe the batch sorting.

In the sentence “In this case, the size distribution of ZnS nanocrystals is still slightly wider, which can be attributed to the difference in the size of SDBS-wrapped reactant package”, it is not clear what ‘wider’ is referring to, making the comparison hard to understand.

ANASWER: Dear reviewer, we wanted to compare the current products with the ideal sample with narrow size distribution. I'm sorry we can't express it clearly. Now we have modified it. In view of the wide particle size distribution of sample, the modification is as follows: “In the present case, the size distribution of ZnS nanocrystals is wide, which can be attributed to the difference in the size of SDBS-wrapped reactant package.”. (On page 6, in lines 148-149)

The terminology referring to the three different samples gets confused e.g. “…the samples Q-ZnS-Iʹ and c displayed…“, also in the materials and methods the labelling refers to a, b, c but it’s not obvious why.

ANASWER: Dear reviewer, the terms for three different samples are unified as Q-ZnS-I, Q-ZnS-Iʹ, and Q-ZnS-Iʹʹ. In lines 77-78, the “(a: 50 oC, b: 100 oC, c: 150 oC)” has been modified to “(Q-ZnS-I: 50 oC, Q-ZnS-Iʹ: 100 oC, Q-ZnS-Iʹʹ: 150 oC)”. Moreover, several other places in the text have also been replaced and modified.

The absorption spectrum shows an excitonic peak at around 269 nm, accompanied by a steamed bun tail.” The UVvis spectrum looks incorrect. I would personally interpret this as a wide excitionic peak around 300 nm, with the peak at ca. 275 nm and the subsequent drop to near zero below that an indication that the cuvette was not suitable for measurements below ca. 290 nm. Were the cuvette and spectrometer the same as the one used in Figure 5a?

ANASWER: Dear reviewer, I'm very sorry for our negligence. Thank you very much for your careful review! In the figure 6a, it is indeed a broad excitation peak at about 300 nm, with a peak of about 275 nm. All UV-Vis absorption measurements in this paper are based on quartz glass absorption cell and the same type of spectrometer. As to the cliff-jumping descent to near zero after 275 nm in Fig. 6c, it may be related to the size distribution of the sample (Fig. 6b') .

On page 9, in lines 234-236, the sentences “The absorption spectrum shows an excitonic peak at around 269 nm, accompanied by a steamed bun tail. This may be due to a slightly wider size distribution of ZnS nanocrystals.” have been modified to “The absorption spectrum shows a broad excitation peak at about 300 nm, with a peak of about 275 nm and the subsequent abrupt descent to near zero. This may be due to a slightly wider size distribution of ZnS nanocrystals of Q-ZnS-Iʹ-ii than ideal sample with single size distribution. The trend of the absorption curve may correspond to the size distribution of the sample (Fig. 6b').”.

Comparing Figure 5b with Figure 6c, there is a very large discrepancy in the spectra for the I’ sample. Figure 5b shows near zero emission intensity at 350 nm, whereas Figure 6c has a distinct peak at 350 nm which is ascribed to band edge emission. Further, the supposed trap state emission at 460 nm in Figure 6c is far higher than the band edge emission, whereas in Figure 5b the trap state is lower in intensity, and also at a much different wavelength position. It is quite confusing, and I think this section needs to some much more careful explanation.

ANASWER: Dear reviewer, thank you very much for your professional review and comment! We would like to present our analysis from the following points. The difference of emission peaks in Fig. 5b and Fig. 6c may be caused by the difference of particle size and particle size distribution. Each emission map reflects the structural characteristics of the main particles in the sample. Fig. 5b reflects wide multiple overlapping emission peaks of the original sample (without particle size optimization). Its particle size distribution is wide and the emission at 350 nm is overlapped in the band-edge emission peaks (about 384 nm) of main particle. It can be concluded that the figure 6c presents emission spectra of nanocrystals with small sizes and narrow particle size distribution. Meanwhile, the trap-state emissions of the original sample and batch-sorting sample at about 460 nm are all stronger than the band-edge emission at 350 nm. The ratio of band-edge emission intensity at 350 nm to trap-state emission intensity at near 460 nm is about 1:5.7. After particle size optimization, the ratio of the intensity of these two emissions changed to about 1:3. This trend may be due to the high defect density of small size nanocrystals causing by the immaturity of them.

On page 8, in line 207, the sentences “Because of the wide size distribution of the nanocrystals, its band-edge emission and trap-state emission peaks overlap, showing a broad hilly emission spectrum. Meanwhile, the intensities of emission patterns presented a high and low value crisscross layout, which may be attributed to the difference of the number of respective nanocrystals with corresponding size and the structural reasons such as the density of defects.” have been added.

 On page 10, in line 239, the sentences “……and stronger emission intensity than band-edge emission at 349 nm. The above characteristics of fluorescence emission peaks reflect the small size and narrow distribution of this batch-sorting sample, and the high defect density of the small size nanocrystals possibly due to their immaturity.” have been added to explain the characteristics of fluorescence emission pattern of Q-ZnS-Iʹ-ii.

Also, the PL in Figure 6b, where is the baseline? It seems to be much higher than it should be?

ANASWER: Dear reviewer, this should be the problem in Figure 6c. Here, the UV-Vis absorption and FL spectra are processed in ‘origin’ software. Due to negligence, the fluorescence spectra are not properly processed. Now, the incorrect image has been replaced (Figure 6, on page 10, in line 240).

Reviewer 3 Report

Detailed experiments should be needed. For example, what solvent did author use for centrifugation?

In XRD data, all of XRD peaks shows right-shifted by increasing aging temperature. Why?

We can not use d of Scherrer’s equation as a size of the nanoparticles before confirming that the nanoparticles have a single crystalline nature. Because d of Scherrer’s equation indicates the size of crystal grain in the particles. Please add the evidence to show that the synthesized ZnS nanoparticles are single crystal. In addition, add size distribution to show size change of the nanoparticles.

In FR-IR, we can see FT-IR peaks of SDBS when we just mix nanoparticles with SDBS. Therefore, it's not a strong evidence to show the attachment of SDBS on the surface of the ZnS nanoparticles. Please add other data to show them.  

Author Response

Comments and Suggestions for Authors and our responses:

Detailed experiments should be needed. For example, what solvent did author use for centrifugation?

ANASWER: Dear reviewers, thank you very much for your professional comments. Blue font is used in the revised part of the text for highlighting.

On page 2, in lines 75-76, the sentence “Finally, the aged masterbatch was centrifugation washed several times and dried at 60 oC to obtain the desired product.” has been modified to “Finally, the aged masterbatch was centrifugation washed with deionized water (three times) and absolute ethanol (one time) respectively, and dried at 60 oC to obtain the desired product.”. 

Moreover, on page 2, in lines 78-81, the descriptions for batch sorting operation “Several batches of ZnS quantum dots with different particle size distributions were separated by adjusting the rotational speed of centrifuge (3000 and 4000 rpm (revolutions per minute), respectively) from the Q-ZnS-Iʹ. The supernatant obtained at 3000 and 4000 rpm of the sample is recorded as the Q-ZnS-Iʹ-i and Q-ZnS-Iʹ-ii.” have been modified to “Several batches of ZnS quantum dots with different particle size distributions will be obtained by adjusting the rotational speed of centrifuge (3000 and 4000 rpm, respectively) from the Q-ZnS-Iʹ. Herein, the redispersed sample Q-ZnS-Iʹ was centrifuged at 3000 and 4000 rpm, respectively, and supernatant and precipitation were separated. The supernatants are labeled Q-ZnS-Iʹ-i and Q-ZnS-Iʹ-ii, respectively.”

In XRD data, all of XRD peaks shows right-shifted by increasing aging temperature. Why?

ANASWER: Comparing the three patterns, we can find that not all peaks in the spectrum of Q-ZnS-Iʹ' have high angular shift, but only a few peaks have high angular shift, which may be a reflection of lattice distortion (anisotropic contraction) caused by macro residual stress, such as twinning and agglomeration structures in Q-ZnS-Iʹʹ. (on page 4, in lines 120-123, a description for the microstructure of Q-ZnS-Iʹʹ has been presented: “It is worthwhile to note that some nanocrystals in Q-ZnS-Iʹʹ generated zigzag twinning structures containing a large number of stacking fault and dislocation, and several nanoparticles formed agglomeration (Supporting Information, Fig. S1).”.

We can not use d of Scherrer’s equation as a size of the nanoparticles before confirming that the nanoparticles have a single crystalline nature. Because d of Scherrer’s equation indicates the size of crystal grain in the particles. Please add the evidence to show that the synthesized ZnS nanoparticles are single crystal. In addition, add size distribution to show size change of the nanoparticles.

ANASWER: Dear reviewers, thank you very much for your professional comments. In view of the slightly poorer crystallinity of our samples and the fact that some of them contain twin particles and aggregate, the use of Shelley's formula is inappropriate. Therefore, we deleted the content of calculation based on XRD, including grain size and lattice contractions. On page 3, in lines 103-110, the sentences “The Scherrer’s equation: d =0.89λ/βcosθ is commonly introduced to calculate the effective average size (d) of nanocrystals with less error, where d is the mean diameter of crystallite, λ is the X-ray wavelength equal to 0.154 nm, β is the full width at half maximum and θ is the half diffraction angle. The calculated result indicated that the average sizes (d) of nanocrystals in the Q-ZnS-Iʹ are 5.2 ± 0.2 nm approximately. The cell constants were calculated to be a = b = c = 0.5388 nm from (111) peaks of Q-ZnS-I after refinement. They revealed that the lattice contractions of Δa = 0.39% occurred against to the reported data (a = b = c = 0.5409 nm), which implied the smaller sizes of ZnS nanoparticles and the high surface to volume ratio of nanocrystals.” have been deleted.

Meanwhile, we added the size distribution plots for the samples Q-ZnS-Iʹ-i (3000 rpm) and Q-ZnS-Iʹ-ii (4000 rpm), and described the particle size distribution.

On page 9, in lines 233-234, the sentence “Figs. 6a and 6b displayed the TEM pictures of the Q-ZnS-Iʹ-i (3000 rpm) and Q-ZnS-Iʹ-ii (4000 rpm), respectively.” has been modified to “Figs. 6a, 6a', 6b and 6b' displayed the TEM pictures and size distribution plots of the Q-ZnS-Iʹ-i (3000 rpm) and Q-ZnS-Iʹ-ii (4000 rpm), respectively.”.

On page 9, in line 227, the sentence “Meanwhile, the size of main nanocrystals in Q-ZnS-Iʹ-ii is smaller than that in Q-ZnS-Iʹ-i.” has been added.

In FR-IR, we can see FT-IR peaks of SDBS when we just mix nanoparticles with SDBS. Therefore, it's not a strong evidence to show the attachment of SDBS on the surface of the ZnS nanoparticles. Please add other data to show them.  

ANASWER: Dear reviewers, the FIIR of SDBS and Q-ZnS-I' sample were carefully compared and analyzed again. It can be found that the FIIR characteristic peaks corresponding to SDBS sulfonic group (< 1250 cm-1, indicated with red arrows) shift or weaken in the FIIR spectrum of Q-ZnS-I' sample. The locally enlarged image shows this more clearly. For example, the peaks at 613 and 841 cm-1 in the FIIR spectra of SDBS, which they respectively respond to in-plane and out-of-plane vibrations of sulfur-oxygen bonds, presented obvious shift and weakening in the FIIR spectra of Q-ZnS-I' sample. Meanwhile, the figure 4 has been replaced by a brighter image.

On page 6, in line 139, the sentence “……, and presents obvious shift and weakening compared with that of SDBS, ……” has been added.

Round 2

Reviewer 1 Report

There are still some mistype errors but the paper can be accepted for publication.

Author Response

ANSWER: Thank you! Dear reviewer, we have continued to revise our manuscripts to improve them.

Reviewer 2 Report

Thank you very much to the authors for their careful edits and responses.

A final comment. I disagree with this statement... "As to the cliff-jumping descent to near zero after 275 nm in Fig. 6c, it may be related to the size distribution of the sample (Fig. 6b')." In my opinion this is a measurement error and not a real result. I think it is sufficient to cut the spectrum at 280 nm, discarding the sharp drop. Then modify the manuscript text. The relevant information is above 280 nm anyway.

Author Response

 ANSWER: Dear reviewer, according to your comments, we have cut the spectrum at 280 nm, discarding the sharp drop and modified the manuscript text. Thank you !

  On page 13, in lines 265-266, the sentence “with a peak of about 275 nm and the subsequent abrupt 265 descent to near zero” has been deleted.

Reviewer 3 Report

Thank you for your fast revision.

I'm still have some questions.

I do not agree the comments for XRD shifting. In Fig. 2, all peaks of 'a' exhibited right shifting compared with peaks of other sample. If author want to say that there is not all right shifting, please show other data. In addition, if author want to describe "zigzag twinning structures containing a large number of stacking fault and dislocation, and several nanoparticles formed agglomeration", please add ref to prove it. Author describe "The broadening of peaks in all XRD spectrums indicates very small size of as-prepared Q-ZnS. But the diffraction peaks from Q-ZnS-I to Q-ZnS-Iʹʹ become sharper and sharper and the half peak width become narrower and narrower, implying that the crystallinity of nanocrystals in the Q-ZnS was improved along the aging temperature." However, XRD peaks in Fig. 2 shows that peaks of 'a' has broad peaks compared with peaks of other samples.

Sorry to border you, but I thinks it is important to improve the quality of this mansucript.

Author Response

ANSWER: Dear reviewer, samples Q-ZnS I, Iʹ, and Iʹʹ have been synthesized again according to the synthetic procedure and XRD tests have been carried out. The original XRD pattern was replaced by new one (Fig. 2).
 1) Comparison with standard diffraction patterns: Firstly, as for the shifting of the peak position, it may be that our expression is not very clear, resulting in some confusion of understanding. What we're saying is that not all right shifts are relative to other samples, not to standard samples. Now, we have listed all the peaks and compared them with the standard peaks. The data show that the diffraction peaks are indeed shifted to the right relative to the standard sample, only the magnitude of the shift is different. The diffraction peak shifts to the right suggested that the lattice shrinkage exists in the sample particles caused by the small sizes of nanocrystals.

 2) Comparison of diffraction spectra among three samples Q-ZnS I, Iʹ, and Iʹʹ: The diffraction peaks of samples I, Iʹ, and Iʹʹ showed their intensity enhanced along the increase of aging temperature (the intensites on the (111) peaks of samples Q-ZnS I, Iʹ, and Iʹʹ are about 202, 286, and 336), implying that the crystallinity of nanocrystals in the Q-ZnS was improved with the increasing of aging temperature. As for the FWHM, XRD calculation shows that there is little difference between the three (the FWHMs on the (111) peaks of samples Q-ZnS I, Iʹ, and Iʹʹ are about 0.766, 0.780, and 0.738). Therefore, the half-peak width value is no longer used as a parameter to measure crystallinity.

  3) Dear reviewer, in our previous report on the synthesis of zinc sulfide (Dalton Trans., 2016, 45, 2409-2416. DOI: 10.1039/c5dt03783b.), we discussed the structure of multiple twinning. Herein, through the solid-state chemical reaction synthesis route, we prepared zizag-type multi-twinning structure again and characterized it by TEM (Supporting Information, Fig. S1). Now, we quoted this literature.

  On page 5, in lines 108-115, the sentences “Three characteristic diffraction peaks at 2θ values of about 28.7, 47.4, and 56.5 to Q-ZnS can be indexed to the crystal planes of (1 1 1), (2 2 0) and (3 1 1) of cubic phase ZnS respectively; which matched with the reported data very well (JCPDS card file No. 65-9585). The broadening of peaks in all XRD spectrums indicates very small size of as-prepared Q-ZnS. But the diffraction peaks from Q-ZnS-I to Q-ZnS-Iʹʹ become sharper and sharper and the half peak width become narrower and narrower, implying that the crystallinity of nanocrystals in the Q-ZnS was improved along the aging temperature.” have been modified to “Three characteristic diffraction peaks at 2θ values of about 28.91/28.82/29.14, 48.23/48.29/48.16, and 57.11/56.8/56.9 to samples I, Iʹ, and Iʹʹ can be indexed to the crystal planes of (1 1 1), (2 2 0) and (3 1 1) of cubic phase ZnS respectively; which matched with the reported data very well (JCPDS card file No. 65-9585, standard values of three diffraction peaks: 28.56, 47.50, and 56.37, respectively). The results showed that the diffraction peak shifts to the right relative to the standard sample in varying degrees, suggesting that the lattice shrinkage exists in the sample particles. At the same time, the diffraction peaks show obvious broadening, suggesting that the sizes of nanocrystals are small, and these small grains will cause lattice shrinkage. On the other hand, the comparison of the diffraction peaks of samples I, Iʹ, and Iʹʹ showed that the intensity of the diffraction peaks enhanced along the increase of aging temperature, implying that the crystallinity of nanocrystals in the Q-ZnS was improved with the increasing of aging temperature.”.

On page 6, in line 127, the phrase “……which are similar to previous reports [21]” has been added.

Round 3

Reviewer 3 Report

The author responded to the given question using sufficient data and references.

Therefore, I think this manuscript is acceptable to Nanomaterials.

Thank you very much.